# Pasting, Rheological, and Tribological Properties of Rice Starch and Oat Flour Mixtures at Different Proportions

**DOI:** 10.3390/foods11142115

**Published:** 2022-07-15

**Authors:** Cunshe Chen, Ping Liu, Jinnuo Cao, Zhixuan Ouyang, Zhihua Pang

**Affiliations:** Beijing Advanced Innovation Center for Food Nutrition and Human Health, Beijing Technology & Business University (BTBU), Beijing 100048, China; chencs@th.btbu.edu.cn (C.C.); liuping970928@163.com (P.L.); jinnuocao@163.com (J.C.); d18801075352@163.com (Z.O.)

**Keywords:** rice starch, oat flour, pasting, rheology, tribology

## Abstract

Rice starch (RS) and oat flour (OF) were mixed in different proportions, and the pasting properties, particle size, rheology, and tribological properties of the mixed system were analyzed. According to the RVA results, OF inhibited the starch pasting, and the pasting temperature and peak viscosity of the mixed system increased. The particle size shifted toward the small particle size after the mixing of RS and OF components, and the RS/OF 9/1 particle size is the smallest. All samples exhibited shear dilution behavior and the viscosity of the system could be significantly increased at a 10 wt% RS content. At sliding speeds of >1 mm/s, the friction of the mixture is usually between the two individual components, which also confirmed the association or interaction between the two polymers.

## 1. Introduction

Oats are widely considered as a quality health food [1,2,3]. It is an excellent source of β-glucan, protein [4], unsaturated fatty acids [5], and unique oat polyphenols with powerful antioxidant activity [6]. Oats are often used in the production of healthy snacks and meals [7] due to valuable health benefits including reducing cholesterol, and managing blood sugar and obesity [8,9]. OF is becoming increasingly popular as an ingredient in the food industry. Studies have proven that OF shows a higher paste viscosity compared to wheat and rye flour [10,11,12]. Information about its pasting properties is essential for the development of new products and the adjustment of processing parameters. Mixing OF with other substances would be an attractive way to develop new products.

Rice is an important cereal that sustains about half of the world’s population. The main component of rice is starch, which accounts for more than 80% of the total composition [13]. RS can be used as a texturizing agent to improve product viscosity, syneresis control, and shelf-life characteristics [14,15,16,17]. It is widely used in rice flour, rice cakes, and various snack foods because of its light color, fine grain, good flavor, and great processing characteristics [18,19]. However, it, in common with other cereal starches, has also negative aspects, such as gel retrogradation and a tendency to produce undesirable weak-bodied, cohesive, rubbery pastes or gels under extended cooking, and high shear or acidic conditions [20].

To address these drawbacks of RS, researchers have blended natural RS with a variety of substances [21,22,23,24]. It was found that the rate and extent of regrowth as well as rheological and structural changes in RS gels were reduced by the addition of β-glucan [25]. Moreover, lipids can interact with starch molecules to form starch–lipid complexes [26], thus affecting the rheological properties of starch [27]. Therefore, a certain amount of oats added to rice starch may build a food system with a specific physical characteristic structure.

The purpose of this study was to investigate the properties resulting from different blending ratios of RS and OF, and to observe the gelatinization, particle size, rheological, and tribological properties of the different systems, which were closely related to the taste of food. To the best of our knowledge, no such results have been reported. This study provided data to support the construction of rice starch–oat flour blended food systems.

## 2. Materials and Methods

### 2.1. Methods

#### 2.1.1. Rice Starch Preparation

The rice powder was obtained by crushing and sieving an appropriate amount of rice (purchased from the local market) through 80-mesh. The rice powder was placed in 0.3 wt% NaOH with a ratio of 1:2, then stirred by electronic stirrer and soaked for 4 h. The mixture was centrifuged at 4000 r/min for 15 min, and the yellow part of the supernatant was removed. We placed the calibrated pH meter probe into the mixture and then adjusted the pH to 6.5–7.0 using 1 mol/L NaOH and 1 mol/L HCl, centrifuged at 4000 r/min for 15 min, and the supernatant and the dark sediments were removed. All the above processes were repeated 3 times. The above precipitate was placed in ethanol solution with a ratio of 1:1 for 2 h, centrifuged at 4000 r/min for 15 min, and the supernatant was removed. Then the pH of the resultant mixture was adjusted to 6.5–7.0, centrifuged at 4000 r/min for 15 min followed by the removal of the supernatant and the dark sediment (repeated 3 times). Then the sediment was oven-dried at 45 °C for 8 h, crushed through 80-mesh sieve, and set aside.

An appropriate amount of crushed oats (purchased from the local market) was taken and sifted through an 80-mesh sieve to make OF.

#### 2.1.2. Sample Preparation

The RS and OF were mixed well at the ratio of 10:0, 9:1, 8:2, 7:3, 6:4, 5:5, 4:6, 3:7, 2:8, 1:9, and 0:10, (*w*/*w*), respectively [28]. Then 10 g of each sample was taken in a beaker and mixed with 100 mL (50 °C) of distilled water, respectively. The samples were heated on a magnetic stirring heater with a heating program of 50–95 °C to simulate the RVA program, as described below. The entire heating process took 30 min. Then the samples were cooled to 25 °C for further analysis.

### 2.2. Pasting Characteristics of the Mixed System

Pasting characteristics of the mixed system were evaluated using a rapid viscosity analyzer (TecMaster RVA, Perten, Sweden). RS was compounded with OF and mixed well. Testing was performed by the methods prescribed by the American Association of Cereal Chemists (AACC) [29,30]. The moisture content of the sample was determined to be approximately 14 % and 3.0 g of the mixture was taken to determine the RVA pasting parameters of the mixtures. The testing procedure is shown in Table 1.

### 2.3. Particle Size Properties of the Mixed System

The particle size distribution of the condensate in the gel was measured using a SOLD-3000 laser diffraction particle size meter. The samples obtained from pretreatment were diluted 10 times with distilled water, then homogenized in a cell homogenizer for about 10 s. After mixing, an appropriate amount of sample was taken immediately in the flow cell for continuous measurement. The particle refractive index was set to 1.5 and the precision was 0.001. The particle size polydispersity was characterized by the span of the size distribution using the following equation [31]:$$Span=\frac{D_{90}-D_{10}}{D_{10}}$$
where *D*_90_, *D*_10_, and *D*_50_ are the volume diameters at 90%, 10%, and 50% of the cumulative size, respectively.

### 2.4. Rheological Properties of the Mixed System

The rheological behavior tests were conducted by using a controlled stress rheometer (TA Instruments, Model DHR-1, Elstree, UK), equipped with cone plate (40 mm, diameter; 2° angle) geometry. Gap size of 57 μm was used. The viscosity was measured over growing shear of 0.01–1000 s^−1^ range at 25 °C.

Data of shear scan curves were analyzed using rheometer analysis software. The resulting downward flow curve was fitted to the Hershel–Bulkley model:

$$\sigma =\sigma_{0}\;+\;K{\dot{\;\gamma }}^{n}$$
where σ = shear stress, σ_0_ = yield stress, k = consistency index, n = flow behavior index, and $\dot{\gamma }=\mathrm{shear}\;\mathrm{rate}.$ Other parameters such as the difference in the area under the two curves (ΔA), as well as apparent viscosity (η_app_) at shear rate of 50 s^−1^ [32,33].

### 2.5. Tribological Properties of the Mixed System

The friction curves were measured using a new hybrid flat plate friction rheometer (TA Instruments, New Castle, DE, USA), and the oral tongue was simulated using 3M Transfer Hole Surgical Tape 1527-2 (3M Medical, West Bloomfield, MI, USA) [34]. A normal force of 2 N was exerted on the sample during oral-to-mouth processing [35]. The tape was cut into squares and taped flat to the base at the time of sample loading, with the excess glued to the outside of the base. The tape was replaced at the end of each measurement and the base was cleaned. Test conditions were: temperature 37 °C, stress 2 N (i.e., pressure 27.83 KPa), sample volume 1.5 mL, and rotate speed 0.01–100 s^−1^. Two replicates were made for each sample, and three parallel replicates each. The tribological data were filtered to remove the data points whose normal error was greater than 5% from the analysis results.

### 2.6. Data Processing

SPSS Statistics software 22.0 (SPSS, Chicago, IL, USA) was used for the analysis of variance and test of significance (*p* < 0.05), and Origin 8.5.1 and excel software were used to draw the graphs.

## 3. Results and Analysis

### 3.1. RVA Gelatinization Characteristics

In the presence of water (the temperature of the whole system was above the gelatinization temperature), the starch particles were swelled, all the molecular order disappeared, and amylose and amylopectin began to exude. At this stage, the particles were susceptible to mechanical stress, i.e., shearing led to particle rupture, dispersion of amylose and amylopectin, and formation of the paste [36]. The RVA characteristics of RS mixed with OF is shown in Figure 1 and Table 2. The RVA showed the change in viscosity of the sample with the temperature during the pasting process. With the addition of RS, the gelatinization temperature decreased and moved closer to the gelatinization temperature of OF as the percentage of RS increased, which was attributed to the OF that may have inhibited the expansion of RS [37]. The peak viscosity increased significantly to 1842 ± 86 cp after 10 wt% of RS was replaced with OF. As the proportion of OF increased, the peak viscosity of the mixed system gradually decreased and tended to the peak viscosity of OF, which may be due to the stacking effect of the higher OF content in the system, resulting in the higher amylose content and, therefore, the viscosity of the pasted product increased [38]. Further increasing the OF concentration could introduce a higher content of protein in the mixture, which could decrease the pasting viscosity of starch. It has been found that the addition of a certain amount of protein to RS decreases the peak and final viscosity, while increasing the gelatinization temperature in a concentration-dependent manner [39,40,41]. It has been demonstrated that increasing the glutelin concentration up to 25 mg/g starch also resulted in near linear decreases in RVA viscosities, while the effect tapered off with further increases in glutelin concentration [39]. Li et al. also reported that the interaction of β-amylase with RS resulted in a significant increase in the crystallinity, gelatinization temperature, and enthalpy of the modified starch [42]. In summary, OF competes with RS for water to delay RS swelling, and increasing the amount of OF may increase the peak viscosity of the mixed system and gelatinization temperature.

### 3.2. Particle Properties

The particle size distribution of the mixed system of RS and OF is shown in Figure 2. The results showed that the particle size of the mixed system was mainly divided into four zones, namely 4–7 μm, 16–19 μm, 30–14 μm, and 150–160 μm. The span value of the paste OF was 1.99 ± 0.43 and the average particle size was 51.54 ± 4.42 μm. The span value of the paste RS was 1.78 ± 0.51 and the mean particle size was 65.98 ± 6.32 μm. The span values of both samples were >1, indicating a wide range of particle size distribution and a certain degree of dispersion. The swelling force and solubility values of all starches gradually increased when the temperature increased [36]. The proteins in OF lose their water film when they are denatured by heat. As a result, protein particles cannot be dissolved with water and are prone to intermolecular collisions and aggregated precipitation, subsequently causing particle enlargement [43]. The particle size results showed that the particle size shifted toward the small particle size after mixing the two components due to the starch swelling force [44], as well as the protein denaturation. This result can also indicate a decrease in the firmness of the pasteurized OF and RS complex gel.

### 3.3. Rheological Properties

The apparent viscosity of the RS and OF mixture was decreased with the increasing shear rate and was thus characterized by shear thinning. As the shear rate increased, the aggregated particles were broken down into smaller particles, resulting in a decrease in viscosity. The particle size of RS became smaller after mixing with OF, and the particle size of the mixture with a small particle size became smaller after shearing, and thus the viscosity subsequently became smaller. A lower viscosity was observed in RS/OF 9:1, which also showed a small particle size. The Hershel–Bulkley model was selected to fit the downward line of the shear scan curve. The model was selected according to our previous publications [45]. The results showed a good fit (R^2^ > 0.99), and the specific values of the rheological parameters obtained is shown in Table 3. As the percentage of OF increased, the mixture presented a higher consistency factor K. The higher the concentration of starch, the larger the value of the flow behavior index n, indicating that the mixture is more pseudoplastic. The results of yield stress were observed to increase and then decreased with the increasing the percentage of RS. Various factors influenced the variation of parameters when RS was combined with OF. On the one hand, the replacement of RS with OF reduced the total starch content of the system due to the varied composition of oats. The proportion of amylose in oat starch was about 26% higher than in RS, and oat starch was hard, rigid, and fragile. The combined effect of both starches (when mixed) influenced the change in properties and parameters. In addition, the lagging ring area (ΔA), which was an indication of structural damage (thixotropy) and network reconstruction during shear, was also calculated. The highest value of ΔA was found for RS/OF 4/6, indicating that this hybrid system was more prone to structural damage under shear stress and more difficult to reconstitute into a coherent network structure after shear.

### 3.4. Tribological Properties

Tribology is becoming a discipline that contributes to the understanding of oral food processing as well as texture and mouthfeel because it includes the rheological properties of the fluid (lubricant), as well as the surface properties of the interacting substrates in relative motion [46]. The friction characteristic curves (characterized by Stribeck curves) of RS and OF mixtures with different mixing ratios are shown in Figure 3. The various mixing ratios of the two components had a significant effect on the friction coefficient. The RS/OF 10/0 showed the classic Stribeck curve over the entire speed range. At the initial velocity, the coefficient of friction μ increased with the increase in velocity. When the velocity reached 1 mm/s, μ stopped increasing, and this area was called the boundary lubrication zone. The friction coefficient in the boundary region is likely to depend more on the size of the particles than on the deformability of the particles [47]. When the sliding speed was >1 mm/s, the starch particles were entrained to the fixture surface and formed a lubricating film, μ decreased with increasing speed, and this region was called the mixed layer zone. We can observe an increasing trend at higher velocities, μ appeared to increase, and this region is known as the hydrodynamic layer zone. At high entrainment velocities, the two contact surfaces were completely separated, large particles could enter the gap, the system viscosity increased, and the friction coefficient increased. In contrast to RS/OF 10/0, RS/OF 0/10 showed an overall rising friction curve as well as a higher coefficient of friction over the entire sliding speed range. When the sliding speed was higher than 300 μm/s, a decrease in the μ of RS/OF 0/10 was observed.

Mixing RS and OF results in friction that lies between their respective component frictions, with no apparent pattern except for the boundary lubrication layer. This may indicate that there was some interaction between the two components, which results in the friction between the individual components after mixing. Specifically, all hybrid systems exhibited the classical Stribeck curve. At RS percentages above 50 wt%, the coefficient of friction showed an overall increasing trend as the percentage of OF increased. When RS was less than 50 wt%, OF was the main body of the mixture. As the percentage of RS increased, the friction coefficient decreased, which was probably due to the lower content of straight-chain starch in RS than in oat starch, and the texture of the product obtained after pasting was softer, so the friction coefficient was lower. Combining the results of rheology, tribology, and particle size, it can be seen that the mixture is softer and smoother in the particle size range of 4–7 μm (RS/OF 9:1) than in other blends. Therefore, in the RS/OF intermix system, an appropriate increase in the proportion of RS was beneficial to improve the smoothness of the mixed system as well as to reduce the particle residue during swallowing.

## 4. Conclusions

In the RS/OF blend system, appropriately increasing the proportion of RS can improve the smoothness of the blend system and reduce the particle residue when swallowing. Increasing the proportion of OF increases the peak viscosity and pasting temperature of the mixed system, and the mixing of the two substances resulted in a smaller particle size. The rheological results showed that the viscosity of the system could be significantly increased at 10 wt% of RS. Tribological results also indicated that RS and OF had significantly different lubricating properties. The friction of a mixture was usually between two separate components, which can also confirm the association or interaction between the two polymers. Therefore, the lubrication characteristics of the mixture can be improved by changing the intermixing ratio of RS/OF.

## Figures and Tables

**Figure 1 foods-11-02115-f001:**
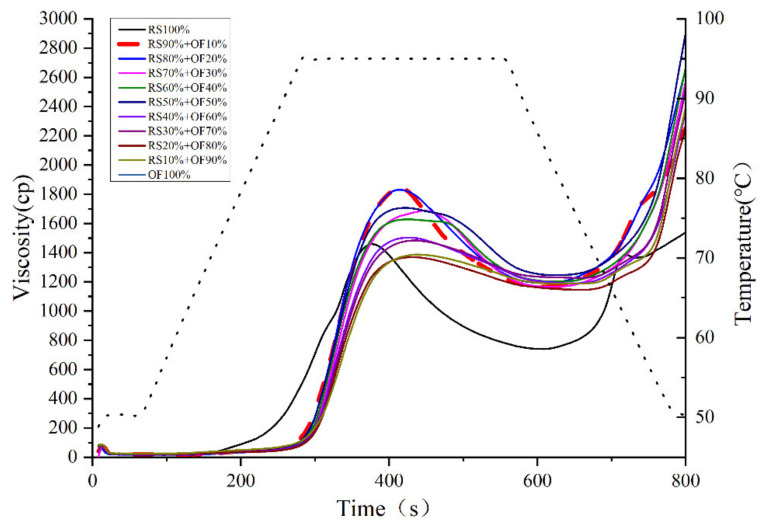
RVA gelatinization curve of rice starch (RS) and oat flour (OF) mixture system.

**Figure 2 foods-11-02115-f002:**
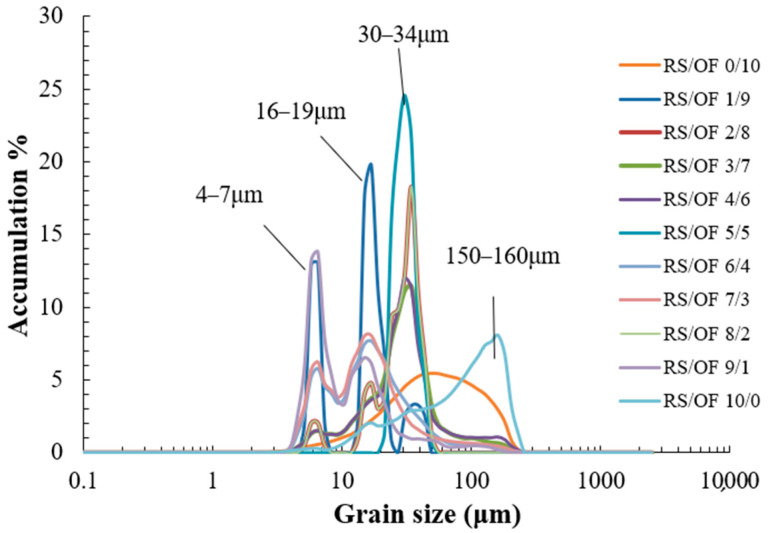
Particle size distribution of rice starch (RS) and oat flour (OF) mixture system.

**Figure 3 foods-11-02115-f003:**
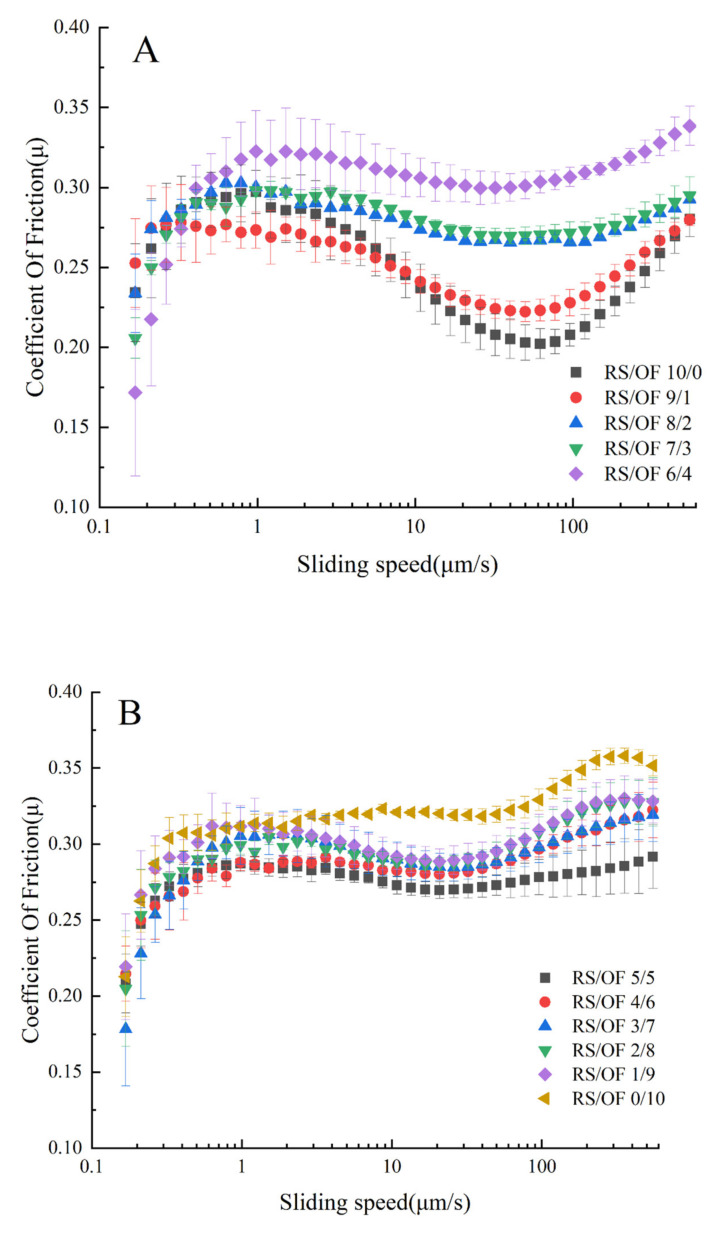
Friction characteristics of rice starch (RS) and oat flour (OF) mixtures with different ratios. (**A**) shows the coefficient of friction when RS is higher than OF. (**B**) shows the coefficient of friction when RS is lower or equal to OF.

**Table 1 foods-11-02115-t001:** **Rapid viscosity analyzer** (RVA) program settings.

Time	Type	Set Value
00:00:00	Temperature	50 °C
00:00:00	Rotate speed	960 r/min
00:00:00	Rotate speed	160 r/min
00:01:00	Temperature	50 °C
00:04:42	Temperature	95 °C
00:07:12	Temperature	95 °C
11:11:00	Temperature	50 °C
00:13:00	End of test	-

**Table 2 foods-11-02115-t002:** RVA gelatinization parameters of rice starch and oat mixture system. RS: rice starch; OF: oat flour.

Sample	Gelatinization Temperature °C	Peak Viscosity cp	Peak Time min	Minimum Viscosity cp	Final Viscosity cp	Attenuation Value cp	Rebound Value cp
RS/OF 10/0	85.55 ± 1.2 ^a^	1461 ± 2 ^c^	6.30 ± 0.05 ^a^	720 ± 6 ^a^	1769 ± 37 ^a^	741 ± 4 ^f^	1028 ± 42 ^a^
RS/OF 9/1	94.13 ± 0.60 ^b^	1842 ± 86 ^g^	6.93 ± 0.00 ^b^	1165 ± 52 ^b^	2929 ± 156 ^e^	677 ± 34 ^e^	1764 ± 209 ^fg^
RS/OF 8/2	94.5 ± 0.00 ^b^	1838 ± 124 ^fg^	6.97 ± 0.14 ^b^	1192 ± 42 ^b^	3347 ± 178 ^fg^	646 ± 82 ^e^	2155 ± 220 ^ij^
RS/OF 7/3	94.75 ± 0.21 ^b^	1682 ± 140 ^d^	7.64 ± 0.05 ^d^	1165 ± 24 ^b^	3282 ± 55 ^f^	517 ± 116 ^de^	2117 ± 31 ^hi^
RS/OF 6/4	94.90 ± 0.57 ^b^	1644 ± 22 ^d^	7.27 ± 0.57 ^c^	1202 ± 36 ^bc^	3206 ± 76 ^f^	443 ± 14 ^d^	2005 ± 112 ^h^
RS/OF 5/5	94.50 ± 1.06 ^b^	1708 ± 86 ^de^	7.00 ± 0.10 ^bc^	1246 ± 52 ^bc^	3231 ± 70 ^f^	463 ± 34 ^d^	1985 ± 18 ^g^
RS/OF 4/6	95.10 ± 0.28 ^bc^	1505 ± 115 ^bc^	7.14 ± 0.09 ^c^	1190 ± 112 ^bd^	2755 ± 321 ^de^	316 ± 1 ^c^	1566 ± 209 ^ef^
RS/OF 3/7	95.03 ± 0.04 ^c^	1485 ± 60 ^ac^	7.24 ± 0.05 ^c^	1223 ± 35 ^c^	2647 ± 11 ^d^	262 ± 25 ^b^	1425 ± 46 ^e^
RS/OF 2/8	94.83 ± 0.11 ^b^	1370 ± 88 ^a^	7.30 ± 0.04 ^d^	1145 ± 38 ^b^	2418 ± 148 ^b^	225 ± 49 ^a^	1273 ± 110 ^b^
RS/OF 1/9	95.05 ± 0.00 ^b^	1480 ± 115 ^ac^	7.37 ± 0.05 ^d^	1256 ± 69 ^cd^	2557 ± 3 ^c^	224 ± 47 ^a^	1302 ± 66 ^c^
RS/OF 0/10	95.35 ± 0.00 ^b^	1435 ± 3 ^ab^	7.00 ± 0.05 ^bc^	1124 ± 24 ^b^	2444 ± 130 ^b^	311 ± 21 ^c^	1320 ± 106 ^cd^

Note: Each group of data was only compared longitudinally, not horizontally; different letters marked represent significant differences between data, *p* < 0.05.

**Table 3 foods-11-02115-t003:** Rheological parameters of rice starch and oat mixture system. RS: rice starch; OF: oat flour.

Sample	Lagging Ring Area ΔA (1/s·Pa)	Yield Stress (Pa)	Consistency Coefficient (Pa·s^n^)	Flow Behavior Index n	Viscosity (at 50 s^−1^) η_50_
RS/OF 0/10	2085 ± 106 ^bc^	9.99 ± 0.59 ^d^	8.38 ± 0.81 ^f^	0.37 ± 0.05 ^a^	1.30 ± 0.22 ^bc^
RS/OF 1/9	1562 ± 100 ^a^	4.70 ± 0.45 ^a^	6.63 ± 0.35 ^e^	0.44 ± 0.01 ^b^	1.37 ± 0.02 ^g^
RS/OF 2/8	2070 ± 189 ^b^	6.42 ± 0.21 ^bc^	3.35 ± 0.08 ^cd^	0.59 ± 0.02 ^c^	1.02 ± 0.06 ^b^
RS/OF 3/7	2051 ± 19 ^b^	7.63 ± 0.73 ^cd^	2.89 ± 0.27 ^bc^	0.60 ± 0.02 ^c^	1.18 ± 0.02 ^cd^
RS/OF 4/6	2390 ± 164 ^cd^	8.69 ± 0.18 ^d^	2.82 ± 0.27 ^b^	0.64 ± 0.01 ^cd^	1.20 ± 0.04 ^e^
RS/OF 5/5	2107 ± 200 ^b^	8.83 ± 0.98 ^cd^	2.67 ± 0.19 ^ab^	0.63 ± 0.01 ^cd^	1.31 ± 0.01 ^f^
RS/OF 6/4	2812 ± 89 ^d^	11.70 ± 0.66 ^e^	3.51 ± 0.52 ^d^	0.64 ± 0.04 ^cd^	1.26 ± 0.07 ^e^
RS/OF 7/3	2159 ± 52 ^bc^	8.31 ± 0.32 ^d^	2.12 ± 0.18 ^a^	0.67 ± 0.01 ^d^	1.12 ± 0.06 ^cd^
RS/OF 8/2	2004 ± 177 ^b^	8.42 ± 0.08 ^d^	2.25 ± 0.12 ^ab^	0.67 ± 0.01 ^d^	1.14 ± 0.06 ^bc^
RS/OF 9/1	1251 ± 103 ^a^	5.37 ± 0.01 ^ab^	2.11 ± 0.09 ^a^	0.65 ± 0.01 ^d^	0.88 ± 0.01 ^a^
RS/OF 10/0	1385 ± 59 ^a^	14.48 ± 0.42 ^f^	4.03 ± 0.16 ^d^	0.68 ± 0.00 ^d^	1.43 ± 0.04 ^g^

Note: Each group of data was only compared longitudinally, not horizontally; different letters marked represent significant differences between data, *p* < 0.05.

## Data Availability

The data presented in this study are available on request from the corresponding author.

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
