# Peer review of "Pasting, Rheological, and Tribological Properties of Rice Starch and Oat Flour Mixtures at Different Proportions"

_foods, 2022, doi:10.3390/foods11142115_

Round 1
Reviewer 1 Report
Your research is very interesting, however, I wish in your conclusion you could have suggested the types of foods which can be formulated with your RS-OF blended flour.

Reviewer 2 Report
In the manuscript, the authors present the Pasting, rheological and tribological properties of rice starch and oat flour mixtures at different 2 proportions. In general, the manuscript has scientific contribution and could be accepted after major revision.
The following are some suggestions for improvements.
Line 61. The last sentence should appear before.
Line 69. The ratio was in weight/weighed (w/w) or other?
Line 72. Added the words “and the mixture was” before “centrifuged at 4000 r/min for 15 min”.
Line 90: Why the authors selected the heating program of 50-95 °C?
Table 1: Why the authors choose the reported RVA program settings (temperature and time)?
Line 123: Use “K” in lowercase.
Line 189: Use the reported values of particle size distribution (D10, D50, and D90) to calculate the Span value (according to Tavares et al. 2021: Microencapsulation of organosulfur compounds from garlic oil using β-cyclodextrin and complex of soy protein isolate and chitosan as wall materials: A comparative study).
Line 206: Why The authors choose the Hershel-Bulkley model to fit the downward line of the shear scan curve, and no other models?
The authors should correlate the results of particle size distribution with the results of rheological properties.
Round 2
Reviewer 2 Report
The authors included all suggestions. The manuscript can be accepted for publication in its current form.